# Alcohol Outlet Density and Deprivation in Six Towns in Bergrivier Municipality before and after Legislative Restrictions

**DOI:** 10.3390/ijerph17030697

**Published:** 2020-01-21

**Authors:** Yasmin Bowers, Adlai Davids, Leslie London

**Affiliations:** 1School of Public Health and Family Medicine, University of Cape Town, Cape Town 7925, South Africa; Leslie.London@uct.ac.za; 2Human Sciences Research Council, Port Elizabeth 6045, South Africa; asdavids@hsrc.ac.za; 3Faculty of Health Sciences, Nelson Mandela University, Port Elizabeth 6031, South Africa

**Keywords:** alcohol outlet density, fetal alcohol syndrome, hot spot analysis, liquor act enforcement

## Abstract

**Introduction**. In 2016, after the Western Cape Liquor Act was enacted, alcohol outlets were mapped in the six towns from a previous 2008 study to determine: (1) alcohol outlet density; (2) the association between deprivation and alcohol outlet density; (3) geospatial trends of alcohol outlet densities; and (4) the impact of alcohol legislation. **Methods**. Latitude and longitude coordinates were collected of legal and illegal alcohol outlets, and alcohol outlet density was calculated for legal, illegal and total alcohol outlets by km2 and per 1000 persons. To determine the impact of legislation, t-tests and hot spot analyses were calculated for both 2008 and 2016 studies. Spearman coefficients estimated the relationship between alcohol outlet density and deprivation. **Results**. Although not statistically significant, the number of alcohol outlets and the density per 1000 population declined by about 12% and 34%, respectively. Illegal outlets were still more likely to be located in more deprived areas, and legal outlets in less deprived areas; and a reduction or addition of a few outlets can change a town’s hot spot status. **Conclusions**. Further studies with larger sample sizes might help to clarify the impacts of the Liquor Act, and the more recent 2017 Alcohol-Related Harms Reduction Policy on alcohol outlet density in the province.

## 1. Introduction

### 1.1. Fetal Alcohol Syndrome in South Africa

The historical trauma of the dop system, whereby the remuneration of farmworkers included alcohol, and the subsequent heavy drinking among sub-segments of the Western Cape Province ‘coloured’ (This paper’s authors do not subscribe to the use of this ’racial’ category) population have resulted in high rates of problem drinking among men and women in the region [1,2]. Despite both media attention and prevention efforts, drinking during pregnancy remains a problem. As a consequence, Fetal Alcohol Syndrome/Partial Fetal Alcohol Syndrome (FAS/PFAS) levels have ranged between 40.5 and 119.4 per 1000 in the Western and Northern Cape Provinces of South Africa [2].

The underlying social determinants of heavy alcohol consumption remain unchanged, and include “poor socio-economic conditions, single-parent families, low levels of maternal education, concomitant use of tobacco and other substances, low religiosity and lack of alternative recreational opportunities” [2]. In addition to these social determinants of alcohol consumption, studies have shown that FAS/PFAS is more common in the rural areas, and communities with the highest published prevalence of FAS/PFAS (Aurora and De Aar) are very isolated geographically [3,4,5].

### 1.2. Alcohol Outlet Density and Alcohol-Related Harms

The relationship between alcohol consumption and health outcomes is complex and dependent upon drinking culture, alcohol regulation and alcoholic beverage quality [6]. Alcohol consumption contributes to over 200 health conditions, including injury and communicable and non-communicable diseases [7].

Alcohol harm increases as the density of alcohol outlets increases, with takeaway chain outlets contributing more substantially to the risk [8]. Therefore, addressing the alcohol outlet density in relation to geography (i.e., per km^2^) or population (i.e., per 1000 persons) is an effective method of reducing alcohol consumption and related harms [9]. The World Health Organization (WHO) recommends reducing the density of retail alcohol outlets to reduce the burden of non-communicable disease [10].

A study in the rural Agincourt sub-district of Mpumalanga Province in South Africa concluded that legal alcohol outlet density was associated with potential problem drinking, but not with heavy drinking for men [11]. One potential explanation for the lack of association between legal village alcohol outlet density and heavy drinking is the availability of home brewed alcohol consumed at traditional and cultural events outside of these taverns and liquor stores [12]. This is a common source of consumption, as the majority of alcohol in Agincourt is sold at informal, unlicensed taverns (i.e., shebeens) [11], a situation commonly reported throughout South Africa [13,14,15].

The issue of outlet density is therefore potentially important in efforts to minimize alcohol-related harm in the community [16]. Determining the density of alcohol outlets (typically per 1000 residents) provides information that can support effective public and community health intervention strategies. Mapping these data can contribute toward the planning of health promotion interventions through the spatial representation of a risk and its relationship to a health outcome [17,18].

### 1.3. South African Policies

In 2015, alcohol was identified as the fifth leading risk factor for death and disability in South Africa. It is also a dominant substance of abuse in the Western Cape [19]. The production and sale of alcohol contributed R93.2 billion to the South African economy (2.9% to the gross domestic product (GDP)), but the cost of alcohol-related harms outweighs the contribution at a net loss R165–236 billion (7–10% of the GDP) [19]. In 2009, the total tangible (healthcare, treatment research and prevention, social and welfare costs, crime, road traffic crashes/damage to motor vehicles) and intangible costs (premature mortality and morbidity, absenteeism, non-financial welfare costs) of alcohol harm to the economy were estimated at 10–12% of the GDP [20].

The burden of disease and the social impact of alcohol misuse led to the Western Cape Provincial government enacting the Western Cape Liquor Act of 2008 (hereafter referred to as the “Liquor Act”), its 2015 amendment, and the 2017 Alcohol-Related Harms Reduction Policy. The Liquor Act increased provision for effective offences and enforcement, and dedicated a fund for combatting the negative social consequences of the liquor trade. [21]. The 2017 Alcohol-Related Harms Reduction Policy has three main parts: (1) enforcement that also brings responsible unlicensed alcohol outlets into the regulated space and the rezoning of outlets in appropriate residential areas; (2) recreational alternatives for high risk individuals and economic alternatives for business owners; and (3) community-based health and social services, including the continuation of FAS education programs, especially on farms [22].

The comprehensive nature of these policies reflects the complex impact that alcohol has on health and well-being in South Africa.

### 1.4. Study Setting

The West Coast district is recognized as an area where FAS is prevalent [23,24]. In 2008, location data of alcohol outlets were collected using a global positioning system (GPS) receiver in six towns of the Bergrivier Municipality in the West Coast district [16]. The alcohol outlet data points were analyzed to determine an association between outlet density and deprivation of an at-risk population susceptible to FAS. Major findings of that study (called Study 1 in this manuscript) concluded that socioeconomic deprivation was associated with a higher concentration of unlicensed outlets and fewer licensed outlets. At the time of Study 1, illegal outlets were predominantly located in more deprived areas, and legal outlets in less deprived areas [16].

Study 1 data were collected before the Western Cape Liquor Act of 2008 (enacted April 2012 and amended in 2015) and will be used as a baseline for comparison to post-Liquor Act alcohol outlet data. In 2016, we repeated the 2008 study by analyzing alcohol outlet data in collected from the same towns to determine statistically significant changes, which could indicate the impact of the Liquor Act in the study area, as well as analyzing the association between alcohol outlet density and deprivation. 

Our 2016 study (called Study 2) had four major objectives: (1) to determine alcohol outlet density (measured as outlets/km^2^ and outlets/1000 persons); (2) to determine the association between deprivation and alcohol outlet density; (3) to describe geospatial trends of alcohol outlet density; and (4) to compare findings of Study 1 with Study 2.

## 2. Materials and Methods 

### 2.1. Data Collection

With the assistance of a local community outreach organizer and local police services, field work was undertaken between June 27th and July 1st 2016 to identify and record legal and illegal alcohol outlets. The same six local towns surveyed in 2008, namely Aurora, Eendekuil, Piketberg, Porterville, Redelinghuys and Velddrif were revisited in 2016.

Recording of legal and illegal alcohol outlets was collected by using the GPS to capture the latitude and longitude (lat/lon) coordinates as points of interest along with their attributes such as legal status, type of legal (i.e., tavern, bottle store, café or restaurant, small grocer, supermarket), type of illegal (i.e., shebeen or shebeen tuck shop) and town name. The lat/lon file of alcohol outlets in the six towns was exported to a comma-separated values (CSV) format and then used to create a shapefile using the Environmental Systems Research Institute (ESRI) ArcMap Geographic Information Systems (GIS) software. GIS is a “framework for gathering, managing, and analyzing data…and organizes layers of information into visualizations using maps” [25].

### 2.2. Alcohol Outlet Densities

Alcohol outlet densities were calculated for legal, illegal and total alcohol outlets using two measures: by km^2^ and by town population. To compare the changes in alcohol outlet density between 2008 and 2016, a chart was created with alcohol outlets/km^2^ and alcohol outlets/1000 persons using population Statistics South Africa (Stats SA) census data. During this process, the 2008 chart data for each town’s area in square kilometers needed to be corrected, and recalculations were made for population density and outlets/km^2^. The percentage of illegal alcohol outlets were also calculated for each town on the understanding that one intent of the Liquor Act was to bring illegal alcohol outlets under the ambit of the Act and to regularize their operations as a form of harm-reduction.

### 2.3. Statistical Analyses

#### 2.3.1. T-tests

To determine the differences in outlet densities between the two studies, t-tests were calculated using an unpaired two-sample two-tailed t-test in Microsoft Excel where alpha is 0.05, which is an equivalent to 95% confidence. T-tests were completed for total, illegal and legal outlets; total, illegal and legal outlets/1000 persons; and total, illegal and legal outlets/km^2^. In Microsoft Excel, the alcohol outlet data were organized by study into two columns. Alcohol outlet data from Study 1 represented the “variable 1 range”, and data from Study 2 represented “variable 2 range”. The null hypothesis for all analyses is “no difference between the two studies”. The alternative hypothesis is “there is a difference between the two studies”.

#### 2.3.2. Spearman Rank Correlation 

Study 2 used the South African Index of Multiple Deprivation (SAIMD) 2011, which is a composite index reflecting four dimensions of deprivation experienced by people in South Africa: income and material deprivation, employment deprivation, education deprivation and living environment deprivation [26]. The SAIMD was based on the 2011 population census. The SAIMD presented the component domains of deprivation at municipal ward, municipality and provincial levels. The SAIMD index was used to rank the municipal wards in the Bergrivier local municipality in terms of deprivation and then correlated with alcohol outlet density found in this study. Because the SAIMD uses wards, alcohol outlet density needed to be calculated for legal, illegal and total alcohol outlets using the ward population published in the ward plans of the municipality as the denominator [27].

There are seven municipal wards in the Bergrivier Municipality, and these were ranked in terms of the SAIMD rankings, with the most deprived ward being ranked as “1”. Alcohol outlet density was expressed as the number of outlets in the three categories (illegal, legal and total alcohol outlets) per 1000 persons. The ward with the highest number of outlets per 1000 persons was ranked as 1, as it had the highest measure of alcohol accessibility. In order to correlate the ranking of deprivation and the ranking of alcohol accessibility, the Spearman Rank Correlation is the most suitable as the distribution of both variables cannot be assumed to be normally distributed [28]. The Spearman Rank Correlation was completed in Microsoft Excel using the Real Statistics Add-in where X = ward local ranking and Y = total/illegal/legal per 1000 ranking.

The SAIMD from Study 1 used geographic datazones for ranking and the SAIMD from Study 2 used municipal wards as mapping units, respectively. Because of this, the mapped SAIMD scores could not be compared between the two datasets. 

#### 2.3.3. Hot Spot Analysis and Deprivation 

Reference data on wards, municipalities, and provinces were obtained from the Municipal Demarcation Board and Research Methodology and Data Centre of the Human Sciences Research Council in South Africa. The shapefile of alcohol outlets in the study area were used in ArcMap software to calculate and display optimized hot spots, which define areas of high and low occurrence. The analysis creates statistically significant hot and cold spots using the Getis-Ord Gi* statistic which evaluates the characteristics of the input feature class to produce optimal results [29].

For ‘Bounding_Polygons_Defining_Where_Incidents_Are_Possible’ the “Wards” polygon layer was chosen. All other optional parameters were left blank. The hot spot results were overlain over the 2011 deprivation rankings by ward. For comparison, both 2008 and 2016 alcohol outlet shapefiles were used to perform an optimized hot spot analysis of the illegal, legal and total alcohol outlets. Study 1’s optimized hot spot analysis results were also displayed over the 2011 SAIMD deprivation ranking by ward, which aided in the comparison process. ESRI further details the optimized hot spot analysis [29].

## 3. Results

### 3.1. Study 1 and Study 2 Data Changes

The total area of Bergrivier is 4407 km^2^ [30], and in 2001 the Bergrivier population totaled 46,334 [31]. In 2011, this same Bergrivier population now totaled 61,897 [30]. This is about a 25% increase over 10 years. Table 1 shows population changes specific to the study area.

Table 2. summarizes the alcohol outlet data collection and density calculations between the two datasets. A map of the study area and the 2016 data collection points can be viewed in Figure 1.

The study area had an overall population increase of an additional 8605 persons, which is a 3.09% annual population growth rate or 35.52% total growth rate between 2001 and 2011. All towns except Redelinghuys had an increase in population. Redelinghuys was the only town to have a decrease in population (−7), and an increase in outlets/1000 persons (+1.8). For the whole study area, there was an overall decrease in alcohol outlets by 11.61% (about 12%). There were four fewer illegal and nine fewer legal outlets. Piketberg had the largest decrease of 14 total outlets (−8 illegal, −6 legal). 

During the data collection process, community members reported an increase in the illegal selling of liquor by using vehicles going directly to farms, which might have influenced the decrease in illegal outlets. Farmers mentioned this as a huge problem, and these vendors overcharge the clients, which further impacts their livelihood. 

### 3.2. Liquor Act Impact

#### 3.2.1. Total, Illegal, and Legal Alcohol Outlets

The total number of alcohol outlets declined 11.61%, illegal declined 8.51%, and legal declined 13.84%; the total number of alcohol outlets per population ratio declined by 34.63%. These differences were not statistically significant.

#### 3.2.2. Alcohol Outlet Accessibility and Density—Outlets/1000 persons

Table 3 summarizes the differences in alcohol accessibility as outlets/1000 persons. Eendekuil had the largest decrease of outlets/1000 persons (from 9.51 to 2.61), and Eendekuil also has a population less than 1000. Only Redelinghuys had an increase of outlets/1000 persons (from 5.17 to 6.97), and Redelinghuys also has a population less than 1000.

In the study area, there are no statistically significant differences in alcohol outlets/1000 persons before and after the Liquor Act (total alcohol outlets/1000 persons *p* = 0.22; illegal alcohol outlets/1000 persons *p* = 0.47; legal alcohol outlets/1000 persons *p* = 0.14).

#### 3.2.3. Alcohol Outlet Accessibility and Density—Outlets/km^2^ persons

Table 4 summarizes the differences in alcohol accessibility as outlets/km^2^. Eendekuil had the largest decrease of total outlets/km^2^ (9.41 to 4.71), and Eendekuil is also less than one km^2^.

There are no statistically significant differences in alcohol outlets/km^2^ before and after the Liquor Act (total alcohol outlets/km^2^
*p* = 0.56; illegal alcohol outlets/km^2^
*p* = 0.68; Legal alcohol outlets/km^2^
*p* = 0.36).

While legal outlets/km^2^ declined in four of the six towns, illegal outlets declined in three towns, and increased in three towns. However, as Table 2 indicates, the percentage of illegal outlets as a proportion of all outlets increased marginally (from 41.96% to 43.43%) for the study area. 

### 3.3. SAIMD and Alcohol Outlet Density

The following Table 5 summarizes the results of the South African Index of Multiple Deprivation (SAIMD) ranking and alcohol outlet/1000 persons by ward.

The towns in our study area that fall within two wards and have two deprivation rankings are Piketberg (wards 3 and 4), Porterville (wards 1 and 2) and Velddrif (wards 6 and 7). In both Porterville and Velddrif, the more deprived wards were more likely to have more illegal alcohol outlets. In Porterville, all eight illegal alcohol outlets were in ward 2, with a SAIMD rank of 1. In Velddrif, all 21 illegal alcohol outlets were in ward 6 with a SAIMD rank of 3.

However, in Piketberg the converse applied, as all nine illegal alcohol outlets were in the lesser deprived ward 4 with a SAIMD rank of 6. Spearman rank correlations of the relationship between SAIMD deprivation ranking and total outlets/1000 persons ranking (Spearman Rho = 0.07 *p* = 0.88), illegal outlets/1000 persons ranking (Spearman Rho = 0.56, *p* = 0.20) and legal outlets/1000 persons ranking (Spearman Rho = -0.64, *p* = 0.12) were not significant.

### 3.4. Maps

The following maps provide spatial detail of alcohol outlet densities within the study area. 

### 3.5. Hot Spot Analyses

Figure 2 and Figure 3 represent optimized hot spot analyses for total alcohol outlets of Study 2 and Study 1, respectively. Figure 4 and Figure 5 represent the optimized hot spot analysis for illegal and legal alcohol outlets of Study 2 and Study 1, respectively. For Study 2, all six towns as had statistically significant hot spots (Figure 2), and the three largest towns had both illegal and legal hot spots—Piketberg, Porterville and Velddrif. The Study 2 illegal and legal hot spot map (Figure 4) displays a demarcation between illegal and legal alcohol outlets along ward/deprivation line in Porterville and Velddrif, where illegal alcohol outlet hot spots extend the most into the more deprived ward, while the legal hot spots extend the most into lesser deprived wards. 

Total hot spot status for Redelinghuys and Aurora changed from “not significant” in Study 1 to a “90% confidence hot spot” in Study 2. For Redelinghuys, there were one illegal and two legal alcohol outlets in 2008, while in 2016, there were two legal and two illegal ones (an increase of one illegal alcohol outlet). In 2008, Aurora had one illegal and two legal alcohol outlets; in 2016, three that were legal. Even though there was no change in the total alcohol outlets for Aurora, the 2016 outlets were more concentrated than in 2008. Due to the low numbers, it is understandable that Aurora and Redelinghuys are not a hot spot for illegal or legal hot spots in either 2008 or 2016. 

In Eendekuil, there were six illegal alcohol outlet points and two legal alcohol outlet points in 2008 but three illegal and one legal alcohol outlet points in 2016. This reduction in three illegal alcohol outlets changed Eendekuil from a 99% confidence illegal alcohol hot spot cluster to a “not significant” area. 

## 4. Discussion

### 4.1. Study 1 and Study 2 differences

The results of Study 2 provide some suggestion that, concomitant with the introduction of the Western Cape Liquor Act in 2008, the alcohol outlet density per population across this rural district declined, as did the absolute numbers of outlets. The lack of statistical significance may reflect the small sample size, and the small number of alcohol outlet data points collected. Nonetheless, the finding that the outlet density per population declined by about a third is consistent with the intent of the Act. What was not found was evidence for a shift in the distribution of outlets from illegal to legal, one of the intentions of the Act.

### 4.2. Alcohol Outlet Density

There was wide variability between towns both in terms of the density of outlets and their absolute numbers. Large towns may be more amenable to licensing and enforcement, while smaller towns may have limited enforcement due to greater distances to local police stations and fewer resources per km^2^. 

Due to the limitations of the study and the lack of the FAS prevalence data, the study is unable to correlate alcohol outlet density (alcohol outlets/1000 persons or alcohol outlets/km^2^) to FAS prevalence. Other studies have used alcohol outlets/km^2^ to predict prevalence of alcohol consumption and examine the association between legal alcohol outlets/km^2^ and potential problem drinking. According to Leslie and colleagues [11] only Velddrif within this study would have a “High”(≥ 1.87/km^2^) legal alcohol outlet density, and may be considered to have an increased risk of potential problem drinking, with a predicted potential problem drinking population prevalence of 27.6% [11].

However, using only legal alcohol outlets and density defined by km^2^ to determine alcohol consumption may not be the most appropriate method, as it does not consider the consumption potential from illegal alcohol outlets (which comprised 43.43% of all outlets in our study) and other means of alcohol production and consumption. Nonetheless, alcohol outlet density is proven to correlate with alcohol consumption, and this information provides insights and value as the study area population is at high risk for FAS and other alcohol-related harms. Fontes and colleagues have pointed to the need for accurate consumption data at household and community levels as important for informing public policy for alcohol harm-reduction, including FAS prevention [33].

It should also be noted that in small, isolated towns, the addition of one outlet can have a significant impact. Leslie and colleagues estimated that prevalence of potential problem drinking associated with a difference of one outlet per square kilometer in all villages was 9.2% [11]. Another study found that living in villages with more alcohol outlets was associated with an increased prevalence of herpes simplex virus 2 (HSV-2) infection in young women, and more specifically, an increase of one alcohol outlet, such as bottle stores or taverns, per village was associated with an 11% odds increase of HSV-2 infection [34].

Our study area was chosen because there are high levels of FAS/PFAS despite the lack of health surveillance data. Collecting alcohol outlet densities before and after the Liquor Act provides valuable health risk data for targeted intervention, education, and enforcement. The aforementioned examples provide confidence in how alcohol outlet density correlates to alcohol consumption and health outcomes. 

### 4.3. Deprivation and Alcohol Outlet Density

Deprivation appeared to be associated with patterns of alcohol outlets, but because of the small study area, the isolated and rural nature of the area, and other competing factors, the Spearman coefficients for this association were not statistically significant. However, in two of the larger towns in the area, Porterville and Velddrif, there was a strong binary association with illegal outlets more common in the more deprived wards. Moreover, there were more total and illegal alcohol outlets/1000 persons in more deprived areas and more legal alcohol outlets/1000 persons in lesser deprived areas. 

### 4.4. Geospatial Trends 

The optimized hot spot analyses supports the literature on the impact that a small number of alcohol outlets have [11,34], especially illegal alcohol outlets in our study. This distinctive trait is what keeps this study conservative when making overarching inferences. In 2016, all six towns had a statistically significant hot spot of total alcohol outlets, even though only three towns maintain statistically significant clusters when separated into illegal and legal. All towns should be viewed at risk for problem drinking or heavy drinking, which has led to a high prevalence of FAS. For example, Aurora has one of the highest documented rates of FAS globally (100 per 1000 persons [2]), yet due to its small size and relatively low amount of alcohol outlet points, it is not considered an illegal or legal hot spot. In these cases, it is useful to identify both illegal and legal alcohol outlets of an area, as the total alcohol outlet hot spot results would be more representative of potential alcohol-related harms. This study supports the usefulness of hotspot analyses in future studies, which may help identify communities at risk where health surveillance data are missing.

### 4.5. Study Limitations

Our results are based upon a small study area and sample size, and it may be that not all illegal alcohol outlet points were collected. However, the methods used in both rounds of data collection were the same, and both surveys collected weekday data so, even if we underestimated illegal outlets, changes over time are not likely to be due to how the data were collected. The study also lacked consumption data and health outcome data (such as FAS prevalence) which meant we were unable to correlate outlet density with health measures. 

## 5. Conclusions

Over the period of eight years, there is some suggestion that there was a decline in overall alcohol outlet numbers and in density per 1000 population in six towns in the Bergrivier Municipality during this period, albeit statistically non-significant. This may reflect the impact of the Western Cape Liquor Act of 2008 and its 2015 Amendment, but many other factors may also be operative. 

The regulatory climate that appears to encourage legal alcohol outlets in lesser deprived areas, and illegal alcohol outlets in more deprived areas appears to have persisted. Given the study area’s at-risk population, there is still need for continued community-level education and intervention, closer attention to the implementation of legislation and evaluating the impact of regulation, such as the Western Cape Liquor Act and the Alcohol-Related Harms Reduction Policy. 

## Figures and Tables

**Figure 1 ijerph-17-00697-f001:**
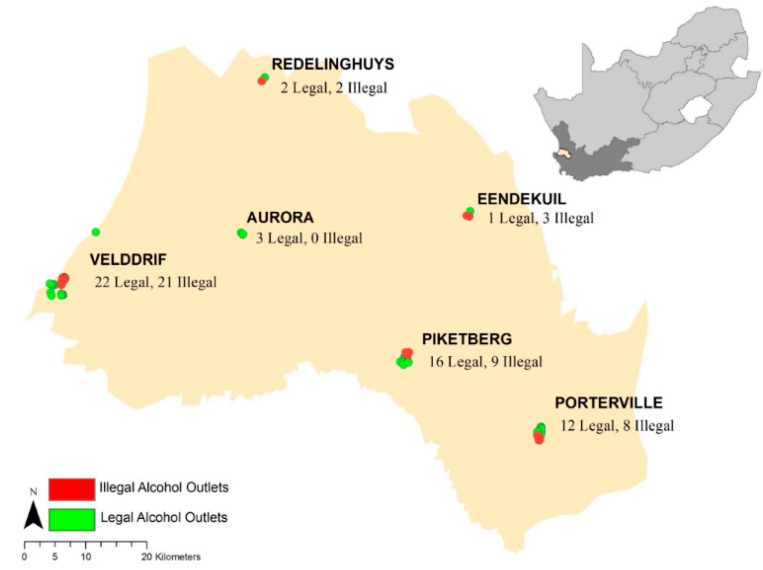
2016 Study Area of collected illegal and legal alcohol outlet points.

**Figure 2 ijerph-17-00697-f002:**
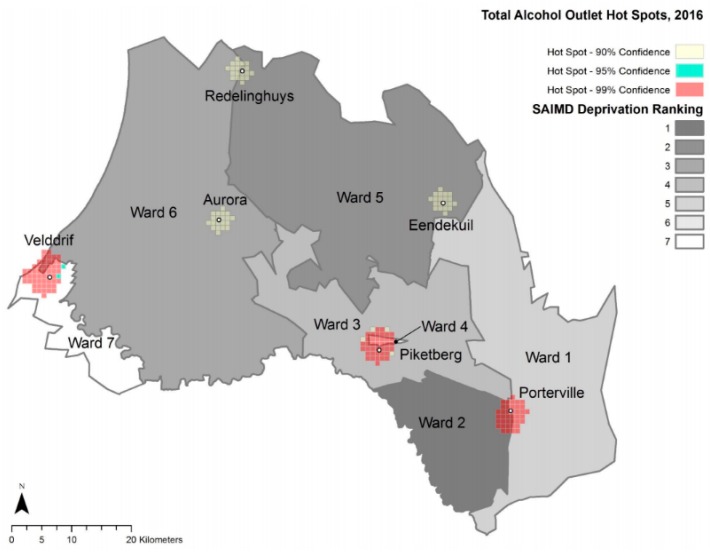
Total Alcohol Outlet Hot Spots, 2016 and Ward Deprivation.

**Figure 3 ijerph-17-00697-f003:**
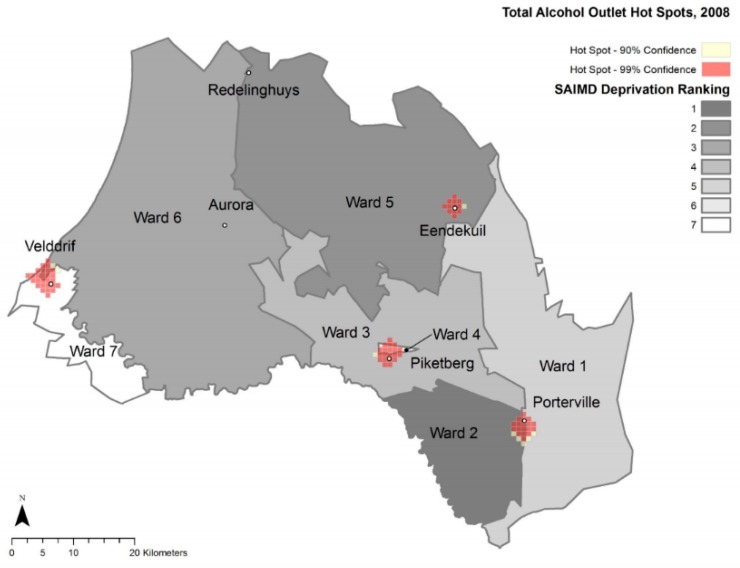
Total Alcohol Outlet Hot Spots, 2008 and Ward Deprivation.

**Figure 4 ijerph-17-00697-f004:**
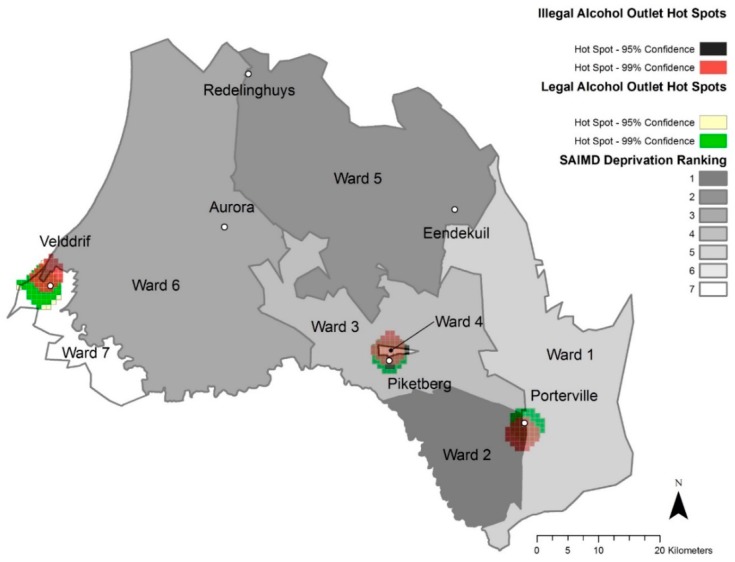
Illegal and Legal Alcohol Outlet Hot Spots, 2016 and Ward Deprivation.

**Figure 5 ijerph-17-00697-f005:**
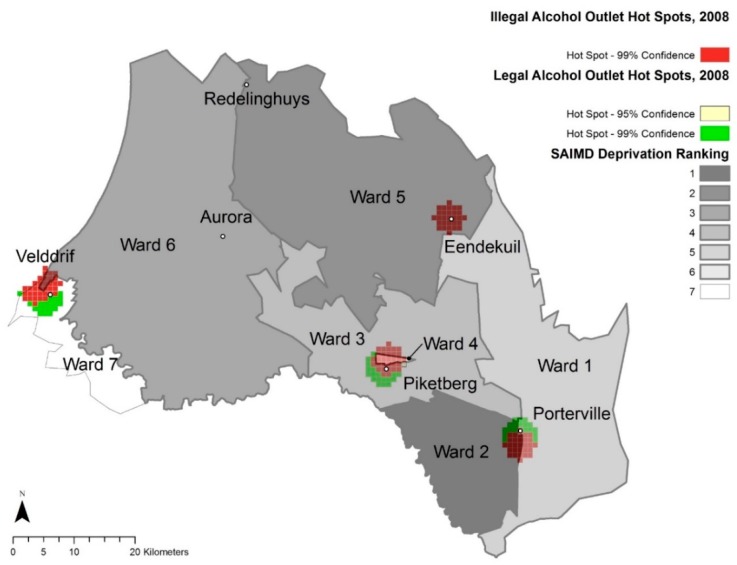
Illegal and Legal Alcohol Outlet Hot Spots, 2008 and Ward Deprivation.

**Table 1 ijerph-17-00697-t001:** Population changes by town.

Town	2001 Total Population [31]	2011 Total Population [31]
Aurora	342	578
Eendekuil	841	1530
Piketberg	9271	12,075
Porterville	5864	7057
Redelinghuys	581	574
Velddrif	7327	11,017
Total Study Area	24,226	32,831

**Table 2 ijerph-17-00697-t002:** Alcohol outlet and density data.

Town	2008 Total Outlets	2016 Total Outlets	2008 Percent Illegal Outlets	2016 Percent Illegal Outlets	2008 Outlets/km^2^ [32]	2016 Outlets/km^2^ [32]	2008 Outlets/1000 Persons	2016 Outlets/1000 Persons
Aurora	3	3	33.33	0	1.84	1.84	8.76	5.19
Eendekuil	8	4	75.00	75.00	9.41	4.69	9.51	2.61
Piketberg	39	25	43.59	36.00	2.94	1.88	4.21	2.07
Porterville	19	20	31.58	40.00	2.38	2.51	3.24	2.83
Redelinghuys	3	4	33.33	50.00	1.42	1.89	5.17	6.97
Velddrif	40	43	40.00	48.84	4.51	4.85	5.46	3.90
Total Study Area	112	99	41.96	43.43	3.23	2.85	4.62	3.02

**Table 3 ijerph-17-00697-t003:** Difference in outlets/1000 persons between two datasets.

Town	Total Outlets/1000 Persons (difference)	Illegal Outlets/1000 (difference)	Legal Outlets/1000 (difference)
Aurora	(−3.57)	(−2.92)	(−0.65)
Eendekuil	(−6.90)	(−5.17)	(−1.73)
Piketberg	(−2.14)	(−1.08)	(−1.04)
Porterville	(−0.41)	(+0.11)	(−0.52)
Redelinghuys	(+1.8)	(+1.76)	(+0.03)
Velddrif	(−1.56)	(−0.27)	(−1.28)

**Table 4 ijerph-17-00697-t004:** Outlets/km^2^ between two datasets.

Town	Total Outlets/km^2^ (difference)	Illegal Outlets/km^2^ (difference)	Legal Outlets/km^2^ (difference)
Aurora	(0)	(−0.61)	(+0.61)
Eendekuil	(−4.72)	(−3.53)	(−1.17)
Piketberg	(−1.06)	(−0.6)	(−0.45)
Porterville	(+0.13)	(+0.25)	(−0.13)
Redelinghuys	(+0.47)	(+0.47)	(0)
Velddrif	(+0.34)	(+0.57)	(−0.23)

**Table 5 ijerph-17-00697-t005:** South African Index of Multiple Deprivation (SAIMD) ranking and ward-level alcohol outlet/1000 persons.

SAIMD Rank ^1^	WARD	Illegal Alcohol Outlets/1000 persons	Legal Alcohol Outlets/1000 persons	Total Alcohol Outlets/1000 persons
1 most deprived	2	1.41	0.35	1.76
2	5	0.35	0.21	0.56
3	6	5.28	1.01	6.29
4	3	0	1.16	1.16
5	1	0	0.87	0.87
6	4	0.87	0.48	1.36
7 least deprived	7	0	1.71	1.71

^1^ Using scores from Noble et al. 2013. The most deprived wards have lower rankings; as the rankings increase, deprivation decreases.

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
