# Peer review of "Alcohol Outlet Density and Deprivation in Six Towns in Bergrivier Municipality before and after Legislative Restrictions"

_ijerph, 2020, doi:10.3390/ijerph17030697_

Round 1

Reviewer 1 Report

Line 78 - "2015 Alcohol-Related Harms Reduction Policy" and then on line 80 - "2017 Alcohol-Related Harms Reduction Policy" ----is this a typo regarding the year (2015/2017) or did it get revised in 2017?  The paper is unclear.

Overall, interesting study.  However, the introduction feels as though the researchers are going to look more into the prevalence of FAS/PFAS.  Once I read the article, it became clear that the researchers were mainly focused on changes associated with alcohol outlet density after policy changes in geographic areas noted to have a high prevalence of FAS/PFAS.  I think the authors need to make it clear that FAS/PFAS prevalence was only used to select the geographic regions rather than discussing the impact on FAS/PFAS prevalence.  

I believe if the authors adjust this concern, it will significantly improve the manuscript.

Reviewer 2 Report

Dear authors,

thank you for this interesting paper evaluating the effect of the 2016 Western Cape Liquor Act and its implication on legal and illegal liquor outlets in six towns of the Western Cape’s Bergrivier Municipality.

General remarks:

Please check your spelling: for numbers, you switch between different notations:

You write 1000 in most cases but for example in line 41, 121, 122 etc. you separate the thousands by comma: 1,000; 46,538 and 61,897.

The same applies for your mentioning of square kilometers: sometimes you write km2 and sometimes km2 Please check all abbreviations for full name explanations. Please check the numbering of your subheadings: line 177: 3.1.2 Alcohol Outlet Accessibility and Density – Outlets/ 1000 persons line 178 3.1.3 Alcohol Outlet Accessibility and Density – Outlets/km2 line 213: 3.3 not 3.2 Maps line 229: 3.4 Hot spot analysis for your results you state p-values. Why do you write “<” and not “=” (see lines 184-186 and lines 193-194) Please restructure your tables 2-4: at present, it is very confusing because some values and columns are named twice in different tables: I would name same parameters for 2008 and 2016 next to each other. For table 2 you could name the absolute values for 2016 as you did for 2008 in column 2-5. For table 3 and table 4 it is then not necessary to name the absolute numbers so that you could state the 2016 differences only. Please change the order of the figures so that matching 2008 and 2016 figures are next to each other and comparable.

Title:

The title implicates that this study takes the prevalence or incidence of the fetal alcohol syndrome in the study area into account. However, in the discussion of the study you write that FAS/FASD prevalence was not correlated with your outcomes concerning liquor outlets.

Instead, I would name the study area and country in which the study was located and please clarify what your study was about (e.g. the Western Cape Liquor Act and density of Liquor stores and its correlation to deprivation/ Comparison to the 2008 study). Otherwise please include data on 2016 FAS/FASD incidence and discuss.

Abstract:

The abstract needs to be restructured and subdivided into “introduction”, “material and methods”, “results” and “conclusion”. Please make clearer what your methods were/ what did you investigate. For example, you could use the aims from line 100-103 for clarification.

Introduction:

Please state your aims (s. materials and methods).

Materials and methods:

Line 100-103: Your aims should be stated at the end of your introduction. Line 103: Reference number 17 does not seem to be the right one here. Please correct or explain. Line 104: Please spell check: June 27th and July 1st, 2016 This might be interesting for a bias section/ the discussion: The time period for the investigation was only 5 days excluding the weekend. Is there a possibility that the number of illegal liquor sales might be different on the weekend? How high would be the bias because of liquor outlets that you did not find? Line 111: What is GIS? Table 1: The first column needs a title (e. g. “Town”) Table 1: Do you have more recent data from 2016? What is the impact of the population growth rate for your study? Did you use the 2011 Census Data for the statistical outcomes, e. g. for outlets/1000 persons or what were your results based on?

Results:

Table 2: please check the title of the last column – did you mean the difference? Otherwise check title of column 5 in table 3. Some columns of the tables are named twice, for example column 2 in table 3 (2008 Outlets/1000 persons) and column 2 in table 4 (2008 Outlets/km2). Table 4: The value for Eendekuil 2016 Outlets/ km2 [difference] differs from the value in Table 3 for the same column title??? Is this correct? Line 197: What is SAIMD? Please explain at first mention. Which references are your 2008 figures based on? Please check lines 239/240: Did you mean more concentrated and did you mean 2008 or 2018?? Do you have any values for FASD/FAS incidence??

Discussion:

This part is not very well structured, no subheadings, please restructure according to your 4 aims as named in lines 100-103. Strengths and limitations/bias are missing As you name FAS in the title of your study it would be nice if you could discuss 2016 values for FAS/FASD incidence with your evaluated alcohol outlet results! You could compare your results to similar studies concerning alcohol outlet density, laws or restrictions in other countries, e.g. Australia, Canada

Reviewer 3 Report

The manuscript “Alcohol outlet density and deprivation in a study area with high prevalence of fetal alcohol syndrome: Changes following restrictive legislation” shows interesting data regarding alcohol outlets in six towns of the Western Cape’s Bergrivier Municipality. Although the title mentions fetal alcohol syndrome, and there is one mention of this syndrome in the abstract, the prevalence of the syndrome itself was not a subject of this study – leading me to suggest it should be removed from the title. In addition, I am unsure regarding the use of the word “deprivation” in the title – what was the Authors’ intention?

In addition, the authors are dealing with very small numbers to be able to spot any statistically significant change. This should be noted in the Discussion. The often use of the word “follow-up” study might not be entirely correct in the case of this study, as it is only a cross sectional study done at two moments. These studies are sometimes called “panel” studies.

The study has, although in Methods, stated the aims, and among those aims there is no connection between FAS and PFAS and alcohol outlet density.

Abstract

The abstract should explain the study more in order. The authors have interesting data which remains hidden by the presentation.

Introduction

The introduction offers a lot of interesting information, but I am not convinced by the use of the term “deprived” and using it as a proxy of FAS/PFAS. The authors should put more effort into explaining and proving this connection in order to use it. In addition, from the title it seems that the authors were to verify that with the reduction of alcohol outlets there will be a reduction in FAS/PFAS, which I cannot see in the paper – and this can be misleading.

Methods

Some parts of the Methods should be in the AIM of the study. In general, this section is well described. A “statistical analysis” section is lacking and it should describe the variables and their roles, as well as statistical analyses used in detail. Hot-spot methodology should be more clear…

Results

Some parts of the Methods (such as Table 1) should be in the Results section. Tables 2 and 3 have some repetition in data which should be avoided, and maybe both tables would be more easily interpreted if there was some graphical representation.

Discussion

The discussion offers some interesting points and comparisons between the data resulting from this study and literature data. The authors again try to link their findings with FAS/PFAS, but the study they have presented actually looks only at correlations between the deprivation index and alcohol outlet density.

Conclusions

The conclusions return to FAS and PFAS as if this was the objective of the study at hand. The conclusions should be re-written to follow the Results and Discussion and not introduce additional information which was not the subject of the study.

Round 2

Reviewer 2 Report

The authors addressed all concerns and questions and the manuscript has improved.

Reviewer 3 Report

No further requirements. Congratulations to the Authors.